# Ruling Out Coronavirus Disease 2019 in Patients with Pneumonia: The Role of Blood Cell Count and Lung Ultrasound

**DOI:** 10.3390/jcm10163481

**Published:** 2021-08-06

**Authors:** Gianni Biolo, Nicola Fiotti, Franco Cominotto, Filippo Giorgio Di Girolamo, Emiliano Panizon, Nicola Altamura, Chiara Casarsa, Alessandro Pipoli, Mauro Giordano, Lucio Torelli, Filippo Mearelli, Pierandrea Vinci

**Affiliations:** 1Clinica Medica, Department of Medical Surgical and Health Sciences, University of Trieste, Strada di Fiume 447, 34149 Trieste, Italy; fiotti@units.it (N.F.); fgdigirolamo@units.it (F.G.D.G.); emiliano.panizon@asugi.sanita.fvg.it (E.P.); nicola.altamura@asugi.sanita.fvg.it (N.A.); chiara.casarsa@asugi.sanita.fvg.it (C.C.); alessandro.pipoli@asugi.sanita.fvg.it (A.P.); filippo.mearelli@asugi.sanita.fvg.it (F.M.); pierandrea.vinci@asugi.sanita.fvg.it (P.V.); 2Emergency Department, ASUGI, Ospedale di Cattinara, Strada di Fiume 447, 34149 Trieste, Italy; franco.cominotto@asugi.sanita.fvg.it; 3Pharmacy Department, ASUGI, Ospedale di Cattinara, Strada di Fiume 447, 34149 Trieste, Italy; 4Department of Advanced Medical and Surgical Sciences, University of Campania, L. Vanvitelli, 80131 Naples, Italy; mauro.giordano@unina.it; 5Biostatistics Unit, Department of Medical Surgical and Health Sciences, University of Trieste, Strada di Fiume 447, 34149 Trieste, Italy; torelli@units.it

**Keywords:** COVID-19, community acquired pneumonia, eosinophils, basophils, lung ultrasound

## Abstract

Coronavirus disease 2019 (COVID-19) is characterized by a distinctive blood leucocyte pattern and B-lines on lung ultrasound (LUS) as marker of alveolar-interstitial syndrome. We aimed to evaluate the accuracy of blood leucocyte count alone or in combination with LUS for COVID-19 diagnosis. We retrospectively enrolled consecutive patients diagnosed with community acquired pneumonia (CAP) at hospital admission to derive and validate cutoff values for blood cell count that could be predictive of COVID-19 before confirmation by the nucleic acid amplification test (NAAT). Cutoff values, generated and confirmed in inception (41/115, positive/negative patients) and validation (100/180, positive/negative patients) cohorts, were ≤17 and ≤10 cells/mm^3^ for basophils and eosinophils, respectively. Basophils and/or eosinophils below cutoff were associated with sensitivity of 98% (95%CI, 94–100) and negative likelihood ratio of 0.04 (95%CI, 0.01–0.11). In a subgroup of 265 subjects, the sensitivity of B-line on LUS was 15% lower (*p* < 0.001) than that of basophils and/or eosinophils below cutoff. The combination of B-lines with basophils and eosinophils below cutoff was associated with a moderate increase of the positive likelihood ratio: 5.0 (95%CI, 3.2–7.7). In conclusion, basophil and eosinophil counts above the generated cutoff virtually rule out COVID-19 in patients with CAP. Our findings can help optimize patient triage pending the NAAT results.

## 1. Introduction

Community-acquired pneumonia (CAP) is one of the most common acute infections requiring hospital admission [1]. The diagnosis of CAP generally requires the demonstration of an infiltrate on the chest radiograph in a patient with a clinically compatible syndrome, i.e., fever, dyspnea, and cough. During the coronavirus disease 2019 (COVID-19) pandemic, all patients with possible CAP are required to be tested for SARS-CoV-2, since CAP and COVID-19 pneumonia can show similar clinical presentations [2,3]. Early COVID-19 recognition is crucial for timely patient isolation to prevent the spread of the disease. A positive nucleic acid amplification test (NAAT) for SARS-CoV-2 generally confirms the diagnosis and no additional diagnostic testing is necessary [4,5]. The time to complete NAATs varies from less than an hour (rapid NAAT) to more than a day [6]. While NAATs for SARS-CoV-2 are expected to be very specific (>99%), reported false-negative NAAT from upper respiratory specimens have ranged between less than 5% up to 40% [7,8]. The pooled sensitivity of initial NAAT in 11 studies was 84% [9]. If the initial test is negative but suspicion for COVID-19 remains high, the test is usually repeated, 24 to 48 hours later, from a specimen obtained by nasopharyngeal swab (NPS) or bronchoalveolar lavage (BAL). Patients awaiting definitive diagnosis pose a major triage challenge in the Emergency Department (ED), due to the contagiousness of COVID-19 and frequent space limitation. Accurate and rapid diagnostic tools would be needed to stratify suspected COVID-19 patients until the definitive confirmation by NAAT.

Thoracic imaging is an important complement to NAAT for diagnosing COVID-19 pneumonia [10]. Computed tomography (CT) is the gold standard method for detection of alveolar-interstitial syndrome, which is the distinctive feature of COVID-19 pneumonia [10]. CT is more sensitive than NAAT for COVID-19 diagnosis but also much less specific [9]. Low-dose non-contrast CT is reliable in detecting COVID-19 pneumonia in daily practice, with a significant reduction in radiation dose [11]. Rapid NAAT and low-dose chest CT can now rapidly and accurately confirm or rule out COVID-19 infection [6,11]. Lung ultrasound (LUS) is also of great interest for rapid COVID-19 screening due to its low cost, widespread availability, ease of use, and reduced risk of spreading the infection [10,11,12,13,14,15]. Single or confluent interstitial artifactual signs, defined as B-lines, represent the alveolar-interstitial syndrome on LUS evaluation [13,14,15]. Although LUS and CT may have comparable diagnostic accuracy [10,14], no chest imaging technique can safely rule out COVID-19.

Immune dysregulation plays a pivotal role in SARS-CoV-2 infection leading to a characteristic pattern of blood leukocytes [16]. Lymphocyte, monocyte, basophil, and eosinophil counts are reduced in proportion to the severity of the disease [17,18,19]. Severe lymphopenia is associate with delayed viral elimination and uncontrolled cytokine production, leading to influx of myeloid cells into the lungs [20,21]. In particular, lung influx of blood eosinophils, mediated by interferon-γ secretion, is the key mechanism of alveolar inflammation [16,22]. Eosinophil counts are related to the outcome of COVID-19 patients [23,24]. Basophil depletion in the acute phase of COVID-19 is associated with downregulation of the antibody response to the virus [16]. Several lines of evidence indicate that such leukocyte pattern represents a distinct cellular signature that could allow COVID-19 pneumonia to be identified among patients with CAP of different etiologies [25].

The main goal of this study was to evaluate the accuracy of performing blood leucocyte count to predict SARS-CoV-2 diagnosis in patients hospitalized for pneumonia not requiring either invasive or non-invasive ventilator support. In addition, the diagnostic accuracy of blood leucocyte count in combination with LUS was evaluated. The results were analyzed according to a retrospective, observational, case-control design during the first (inception cohort) and second (validation cohort) pandemic waves.

## 2. Materials and Methods

We retrospectively reviewed all consecutive patients hospitalized from 14 March to 30 April 2020 (*n* = 201) and from 1 September to 31 December 2020 (*n* = 469) in the Academic Unit of Internal Medicine of the University Hospital of Trieste, Italy, admitting patients with or without COVID-19. After a first evaluation at the ED, patients with fever or symptoms and signs of respiratory tract infection, not requiring either invasive or non-invasive ventilator support, were admitted to the internal medicine ward. We enrolled in the study all patients who had evidence of community-acquired, acute, low-respiratory tract infection during the first (inception cohort, *n* = 156) and second (validation cohort, *n* = 280) evaluation periods. Clinical diagnosis of pneumonia was defined as cough or sputum production, chest pain, dyspnea, tachypnea, abnormal lung examination, or respiratory failure, in combination with evidence of pneumonia at chest radiography obtained within 48 hours after emergency department admission. All laboratory data were available within one day from admission. Patients were excluded if they had a clear alternative diagnosis. Two physicians (G.B. and P.V.) retrospectively and independently evaluated all available records from clinical workups and established the diagnosis of CAP or non-pneumonia disease. When the two physicians were in agreement the diagnosis was considered definitive. In cases of disagreement, a third physician (F.M.) established independently the final diagnosis. At ED admission, LUS was performed with a multiprobe machine (MyLab25 Gold-Esaote, Genoa, Italy) whenever an attending physician, who had completed a LUS training course accredited by the Italian Society of Emergency Medicine, was available. LUS was performed in 88 out of 156 and in 177 out of 280 patients with pneumonia in the inception and validation cohort, respectively. Both the anterior and posterior chest were studied with a 4- to 8-MHz linear probe and a 2.5- to 3.5-MHz curved array probe [26]. In agreement with previously published descriptions [12,15], the presence of focal or diffuse interstitial syndrome and/or parenchymal consolidations was considered suggestive of SARS-CoV-2-related pneumonia. Alveolar-interstitial syndrome was defined as the presence of three or more B-lines in a longitudinal plane between two ribs in a given lung region [13]. Anterolateral and posterior chest scans were performed with both convex and linear probes. LUS was performed on the whole thorax by placing the probe in the intercostal spaces. The probe was positioned both perpendicular to the ribs and along the intercostal spaces. Each hemithorax was divided into five areas: two anterior, two lateral, one posterior. The anterior chest-wall, bounded by the parasternal line and the anterior axillary line, was divided into an upper region (from the clavicle to the third intercostal space) and a lower region (from the fourth intercostal space to the diaphragm). The lateral area, between the anterior and posterior axillary lines, was divided into upper and lower halves. Finally, the posterior area from the posterior axillary line to the paravertebral line was identified. For all patients, SARS-CoV-2 detection was at first performed on NPS samples. All NAAT positive cases were considered true positive. Patients with an initially negative SARS-CoV-2 NAAT were classified as COVID-19 positive if a second NAAT (performed within 72 h from the initial assessment), either by NPS or BAL, resulted in being positive. Conversely, patients with at least two SARS-CoV-2 NAAT concordantly negative were classified as true COVID-19 negative cases. The patients epidemiological, clinical, laboratory, and radiological findings were collected and analyzed. Comorbid conditions were evaluated with use of the Charlson Comorbidity Index [27]. Severity of CAP hospital presentation was defined according to the five-point score CURB-65, based on confusion, blood urea nitrogen, respiratory rate, blood pressure, and age [28]. The study complies with the Helsinki declaration and was approved by the institutional review board of the Friuli-Venezia Giulia region, Italy (CEUR). 

### Statistics

The primary end point of the study was the overall diagnostic accuracy, i.e., the percentage of correctly diagnosed patients, from blood cell counts and alveolar-interstitial syndrome by LUS in the differentiation of SARS-CoV-2 positive and negative patients with pneumonia. Results were statistically analyzed according to a retrospective, observational, case-control design in inception and validation cohorts. Mortality was recorded at 10 and 30 days after hospital admission. Continuous variables were expressed as medians and interquartile ranges (IQR). Categorical variables were summarized as counts and percentages. Results of COVID-19 and non-COVID-19 patients were compared by the Mann-Whitney test or the chi-square test, when appropriate. In order to observe the association between blood cell count and COVID-19 pneumonia, binary logistic regression analysis and Receiver Operator Characteristic (ROC) curve were carried out, then the Youden index was used to dichotomize the continuous variables. Cutoff values were generated in the inception cohort and validated in the validation and combined cohorts. To assess the diagnostic performance of dichotomous variables, i.e., blood cell count below cutoff and B-lines in LUS, the results were tabulated in 2 × 2 contingency tables with groups of subjects divided according to positive or negative SARS-CoV-2 NAAT in columns and categories according to test results in rows. Sensitivity, negative predictive values (NPV), and negative likelihood ratios (NLR) were calculated to assess the ability of blood cell count to rule out COVID-19 infection. Specificity, positive predictive values (PPV) and positive likelihood ratios (PLR) were calculated to assess the ability of blood cell count to rule in COVID-19 infection. The diagnostic performance of eosinophil or basophil count below the cutoff and of B-lines with LUS were assessed both individually and in various combinations. The diagnostic performance of the different tests was compared in combined cohort. The McNemar test was used to compare sensitivity and specificity. To compare the NPV and PPV values we used the “N-1” Chi-squared test. All the statistics were obtained by SPSS (Version 21-0, SPSS Inc, Chicago, IL, USA) and MedCalc statistical software (available at https://www.medcalc.org/, last access 4 August 2021).

## 3. Results

A flow chart of enrollment and adjudication in the inception and validation cohorts is shown in Figure 1. Considering the whole population (i.e., inception and validation cohorts), 78% and 60% had clinical and radiological signs of pneumonia, respectively. The causes of hospitalization of the 234 patients who were excluded from the analysis in the inception and validation cohorts were as follows: heart failure, 22%; urinary tract infection and acute kidney injury, 16%; gastrointestinal bleeding, 7%; decompensated liver cirrhosis, 7%; acute exacerbation of chronic obstructive pulmonary disease, 6%; lung cancer, 5%; abdominal infection, 4%; anemia, 5%; trauma, 4%; acute coronary syndrome, 3%; stroke, 3%; atrial fibrillation, 3%; pulmonary embolism, 3%; connective tissue disease, 2%; pancreatitis, 2%; other diagnoses, 7%. Twenty-eight of the 234 excluded patients (12%) were SARS-CoV-2 NAAT positive but they did not have clinical and radiological signs of pneumonia.

Table 1 shows clinical characteristics and outcomes of patients with pneumonia in the inception and validation cohorts with or without positive SARS-CoV-2 NAAT. The Charlson Comorbidity Index and CURB-65 scores were significantly greater in the inception as compared to the validation cohort while 30-day mortality was significantly lower in the validation cohort. Prevalence of COVID-19 positive subjects was lower (*p* < 0.05) in the inception (26%) than in the validation (36%) cohort. In the validation cohort, median age of COVID-19 patients was lower than that of non-COVID-19 patients while 30-day mortality of COVID-19 patients was higher than that of non-COVID-19 cases in the validation but not in the inception cohort. Cell blood count analysis showed significantly lower lymphocyte, monocyte, eosinophil, and basophil counts in the COVID-19 patients. Among patients with COVID-19, 82% had zero eosinophils at presentation, compared to 29% of non COVID-19 patients (*p* < 0.05). Median basophil count was 68% lower in the COVID-19 as compared to that of non COVID-19 patients (*p* < 0.05).

Results of binary logistic regression analysis indicated (coefficient ± SE) that only eosinophils (−0.013 ± 0.005, *p* = 0.009) and basophils (−0.026 ± 0.011, *p* = 0.016) were independently associated with COVID-19 diagnosis. Therefore, we restricted the analysis on susceptibility to COVID-19 pneumonia to eosinophils and basophils only. ROCs for basophils and eosinophils in diagnosis of COVID-19 pneumonia in inception, validation, and combined cohorts are shown in Figure 2, and their AUC in Table 2. The AUCs were highly significant for all cell types in all cohorts. Optimal cutoff values for basophil and eosinophil count generated in the inception cohort were confirmed in the validation cohort (Table 2). 

Table 3 shows the results of diagnostic performance of eosinophil or basophil count below the generated cutoff values, both individually and in various combinations, in inception, validation, and combined cohorts. The diagnostic performance of the different tests was compared in combined cohorts (All). Regarding COVID-19 ruling out, NPV and sensitivity of eosinophils below the cutoff were significantly greater than those of basophils. In the combined cohort, the finding of at least one or both values of basophils and eosinophils below the cutoff, tested as dichotomous variables for COVID-19 diagnosis, was associated with 36% and 7% greater (*p* < 0.001) sensitivity than those of basophils and eosinophils, respectively. According to NPV and NLR, a negative test, i.e., basophils and/or eosinophils above cutoff, was associated with very low probability of SARS-CoV-2 infection and allowed negative from positive COVID-19 patients with pneumonia to be distinguished in about 98% of cases. Regarding COVID-19 ruling in, the finding of both values of basophils and eosinophils below the cutoff, tested as dichotomous variables for COVID-19 diagnosis, was associated with 8% and 36% greater specificity than those of basophils and eosinophils, respectively. According to PPV and PLR, a positive test, i.e., basophils and eosinophils below cutoff, was associated with moderate probability of SARS-CoV-2 infection and allowed positive from negative COVID-19 patients with pneumonia to be distinguished in about 64% of cases.

The prevalence of COVID-19 in the subgroup of patients in whom LUS was performed was 34%. Table 4 shows the results of diagnostic performance of detection of B-lines with LUS alone and in combination with eosinophil or basophil count below cutoff. Sensitivity of LUS was 15% lower than that of basophils and/or eosinophils below cutoff. The NPV of B-lines was significantly lower than that of basophils and/or eosinophils below cutoff. Positive LUS and/or basophils and/or eosinophils below cutoff were associated with 100% NPV and 100% (95%CI, 96–100) sensitivity. Specificity, PPV, and PLR of positive LUS indicate the poor ability of ultrasound to distinguish positive from negative COVID-19 patients with pneumonia. Nonetheless, the combination of B-lines in LUS with basophils and eosinophils below cutoff was associated with about 88% specificity and the ability to distinguish positive from negative COVID-19 patients with pneumonia in about 72% of cases of positive test. Such PPV of positive LUS in combination with basophils and eosinophils below cutoff was not significantly greater than that of basophils and eosinophils below cutoff alone.

## 4. Discussion

Using retrospective data from 436 CAP patients with suspected COVID-19, enrolled during the first and the second waves of the COVID-19 pandemic, we derived and validated optimal cutoff values of basophil and eosinophil count to discriminate SARS-CoV-2 positive from negative patients. COVID-19 was confirmed in 32% of patients by NAAT. The identified cutoffs were ≤17 cells/mm^3^ for basophils and ≤10 cells/mm^3^ for eosinophils. The diagnostic performance of observing basophils and/or eosinophils below the cutoff indicated NPV and NLR of about 98% and 0.04, respectively. Thus, in the 40% of our CAP patients, who had both basophil and eosinophil values higher than the cutoff, SARS-CoV-2 infection could be virtually ruled out before obtaining the NAAT results.

We observed severe eosinopenia and basopenia in our patients with SARS-CoV-2 pneumonia at hospital admission. Our results were in agreement with previous observations in COVID-19 patients requiring hospitalization in acute care settings of different intensity, i.e., from internal medicine wards to intensive care units [17,18,19,20,25]. Regarding eosinopenia, as a simple biological marker of SARS-CoV-2 infection, our results and those of a recent published study in hospitalized patients with pneumonia indicate the same cutoff of 10 eosinophils/mm^3^ and the same sensitivity of 89% for COVID-19 diagnosis [29].

In a subgroup of 265 subjects, with COVID-19 prevalence of 34%, point-of-care LUS was performed to detect focal or diffuse B-lines with or without consolidation as a characteristic pattern of COVID-19 interstitial pneumonia. In agreement with a recent meta-analysis [9], we found that detection of B-lines in LUS is sensitive (86%) but not specific (44%) for the diagnosis of COVID-19. Detection of parenchymal consolidation is an expression of a more advanced stage of the disease [15]. When our analysis included in addition those patients who presented consolidations without B-lines, the diagnostic accuracy of LUS decreased with a slight change in sensitivity (88%; 95%CI, 79–94) but with significant reduction in specificity (24%; 95%CI, 18–31). In our study the diagnostic accuracy of basophils and eosinophils to rule out COVID-19 was much greater than that of LUS. In fact, basophils and/or eosinophils below thresholds as compared to B-line detection with LUS were associated with about 15% greater sensitivity and much lower NLR. Nonetheless, the integration of clinical evaluation with LUS can be used as pre-test probability of suspected COVID-19 pending the results of blood leukocyte count [30].

While the detection of high values of basophils and eosinophils virtually excludes COVID-19, this approach based on circulating white blood cells yields mediocre results in identifying positive patients. Detection of basophils and eosinophils below the cutoff was associated with small increase in the likelihood of disease. Nonetheless, despite the fact that LUS alone has poor specificity for COVID-19 diagnosis, B-lines on LUS in combination with basophils and eosinophils below the threshold are associated with a moderate increase in the likelihood of disease with specificity of 88%. 

This study has several limitations. First, we investigated the diagnostic performance of the combination of basophils and eosinophils to identify SARS-CoV-2 infection in hospitalized patients with severe CAP within 48 h of admission to the ED. It would be necessary to confirm whether the diagnostic criteria we have highlighted are also valid in asymptomatic patients or in those with mild disease [31]. Second, the etiology of non-COVID-19 pneumonia was not systematically determined by microbiological and serological evaluations, unless required by clinical needs, e.g., when empiric antibiotic therapy was not successful. We avoided induced sputum sampling to minimize the generation of potentially infectious aerosols. Evidence shows that basopenia and eosinopenia are distinctive features of severe COVID-19 [16,17,18,19,20,21,22,23,24,25,29]. In bacterial pneumonia the counts of these cells are generally not reduced [32], while pneumonia due to influenza virus is characterized by a tendency to reduce the levels of basophils and eosinophils but not as markedly as in SARS-Cov-19 infection [33,34]. Third, the mortality rate of COVID-19 patients was about 2-times greater in the inception than that in the validation cohort. These differences in mortality were mainly due to a higher Charlson Comorbidity Index and a greater CURB-65 pneumonia severity score in the inception cohort. Inception cohort patients were enrolled during the first COVID-19 pandemic wave, while validation cohort patients were enrolled during the second. Similar changes in mortality between the first and second wave have been described in different geographical settings [35,36,37]. Possible additional explanations include the increased ability to perform an early etiological diagnosis and the availability of randomized clinical trials showing the effectiveness of pharmacological interventions in COVID-19 pneumonia [38,39]. Nonetheless, despite differences in patient severity in the two cohorts, the diagnostic cutoff generated in the inception cohort and the observed diagnostic accuracy of eosinophils and basophils were fully confirmed in the validation cohort.

In conclusion, assessment of blood leukocyte counts at hospital admission in patients with CAP virtually rule out COVID-19 in those cases with basophils and eosinophils greater than 17 and 10 cells/mm^3^, respectively. Our findings will help to optimize the triage of patients with pneumonia pending the results of NAAT and they will also help to improve the management of false negative results on the first NPS.

## Figures and Tables

**Figure 1 jcm-10-03481-f001:**
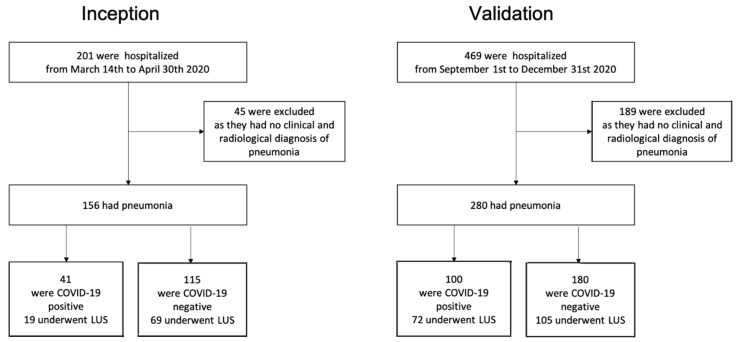
Enrolment and adjudication. LUS, lung ultrasound.

**Figure 2 jcm-10-03481-f002:**
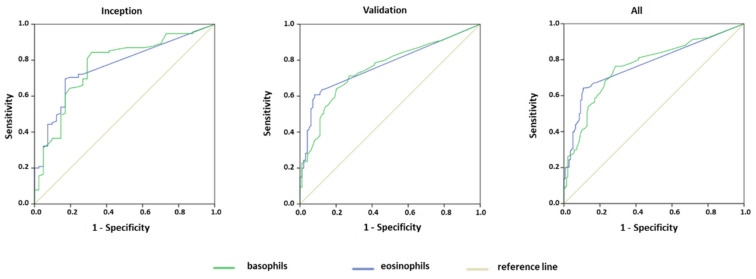
Receiver-Operating-Characteristic Curves for low eosinophil or basophil counts predicting COVID-19 diagnosis in patients with pneumonia in the inception (41 and 115 COVID-19 positive and negative patients, respectively) validation (100 and 180 COVID-19 positive and negative patients, respectively) and combined (141 and 295 COVID-19 positive and negative patients, respectively) cohorts. AUCs and cutoff are shown in Table 2.

**Table 1 jcm-10-03481-t001:** Clinical characteristics and laboratory findings and outcomes of COVID-19 and non-COVID-19 patients with pneumonia in the inception and validation cohorts.

Inception Cohort	All (*n* = 156)	non-COVID-19 (*n* = 115)	COVID-19 (*n* = 41)
Age, median (IQR)—years	81 (73–88)	82 (73–88)	80 (74–86)
Female sex—no. (%)	69 (44)	49 (43)	20 (49)
Charlson Comorbidity Index	3 (1–4)	3 (1–4)	3 (2–5)
CURB-65	3 (2–3)	3 (2–4)	3 (2–3)
10-day mortality, no. (%)	26 (17)	20 (17)	6 (15)
30-day mortality, no. (%)	49 (31)	30 (26)	19 (46) *
C-reactive protein, mg/L	79 (34–165)	78 (37–163)	79 (20–167)
Creatinine	0.98 (0.79–1.45)	0.99 (0.79–1.45)	0.98 (0.80–1.34)
Bilirubine	0.84 (0.57–1.23)	0.93 (0.60–1.29)	0.73 (0.46–1.06)
Hemoglobin, g/dL	11.7 (10.4–13.2)	11.7 (10.5–13.0)	12.3 (10.5–13.9)
Platelet count, per mm^3^	224,000 (178,000–304,000)	218,000 (172,000–302,000)	242,000 (184,000–321,000)
White-cell count, per mm^3^	9485 (7155–12,978)	9550 (7340–12,770)	9120 (6260–14,310)
Neutrophil count, per mm^3^	7159 (5078–10,167)	7146 (5197–10212)	7366 (4919–9673)
Lymphocyte count, per mm^3^	1044 (659–1656)	1138 (682–1848)	914 (615–1216) *
Monocyte count, per mm^3^	683 (465–916)	762 (509–1035)	524 (299–652) *
Eosinophil count, per mm^3^	19 (0–108)	43 (0–135)	0 (0–2) *
Basophil count, per mm^3^	31 (12–48)	38 (21–51)	12 (7–29) *
**Validation Cohort**	**All** **(** ***n*** **= 280)**	**non-COVID-19** **(** ***n*** **= 180)**	**COVID-19** **(** ***n*** **= 100)**
Age, median (IQR)—years	80 (71–87)	81 (74–89)	76 (63–85) *
Female sex—no. (%)	124 (44)	84 (47)	40 (40)
Charlson Comorbidity Index	2 (1–3) ^†^	2 (1–3)	2 (1–3)
CURB-65	2 (1–3) ^†^	2 (1–3)	2 (1–3)
10-day mortality, no. (%)	24 (9)	17 (9)	7 (7)
30-day mortality, no. (%)	62 (22) ^†^	41 (23)	21 (21)
C-reactive protein, mg/L	72 (27–140)	87 (26–176)	60 (31–116)
Creatinine, mg/dL	0.99 (0.76–1.36)	0.98 (0.74–1.34)	1.01 (0.80–1.37)
Bilirubine, mg/dL	0.69 (0.52–1.08) ^†^	0.78 (0.55–1.18)	0.63 (0.49–0.81) *
Hemoglobin, g/dL	12.3 (10.5–13.9)	11.8 (10.2–13.2)	13.5 (11.6–14.9) *
Platelet count, per mm^3^	217,000 (163,000–286,000)	230,000 (167,000–286,000)	201,000 (158,000–284,000)
White-cell count, per mm^3^	8780 (6795–12,635)	9225 (6875–13,155)	8540 (5390–11,410)
Neutrophil count, per mm^3^	7169 (4796–10,651)	7107 (4988–11,145)	6918 (4192–9457)
Lymphocyte count, per mm^3^	957 (569–1359) ^†^	1084 (670–1477)	740 (490–1036) *
Monocyte count, per mm^3^	592 (352–836) ^†^	681 (450–940)	474 (250–628) *
Eosinophil count, per mm^3^	0 (0–70) ^†^	23 (0–133)	0 (0–0) *
Basophil count, per mm^3^	20 (10–39) ^†^	31 (14–51)	11 (6–19) *

^†^, *p* < 0.05; validation vs. inception cohort. *, *p* < 0.05; COVID-19 vs. non-COVID-19.

**Table 2 jcm-10-03481-t002:** Generation of optimal cutoff in inception, validation, and combined cohorts.

		Positive/Negative	AUC	CI 95%	*p*	Cuoffs
Basophils	Inception	41/115	0.77	0.68–0.86	<0.0001	17.5
	Validation	100/180	0.75	0.70–0.81	<0.0001	17.5
	All	141/295	0.76	0.72–0.81	<0.0001	17.5
Eosinophils	Inception	41/115	0.77	0.68–0.84	<0.0001	10.5
	Validation	100/180	0.77	0.72–0.83	<0.0001	10.5
	All	141/295	0.77	0.73–0.82	<0.0001	10.5

**Table 3 jcm-10-03481-t003:** Diagnostic performance of eosinophils and basophils below cutoff in the diagnosis of COVID-19 pneumonia.

		Sensitivity	Specificity	PPV	NPV	PLR	NLR
Basophils ≤ 17 cells/mm^3^	Inception	68 (52–82)	84 (76–90)	61 (49–71)	88 (83–92)	4.36 (2.72–7.00)	0.38 (0.24–0.59)
	Validation	73 (63–81)	72 (65–78)	59 (52–65)	83 (77–87)	2.58 (1.98–3.35)	0.38 (0.27–0.53)
	All	72 (63–79)	77 (71–81)	59 (54–65)	85 (81–88) ^a,b^	3.06 (2.43–3.86)	0.37 (0.28–0.48)
Eosinophils ≤ 10 cells/mm^3^	Inception	81 (65–91)	70 (60–78)	49 (41–56)	91 (84–95)	2.72 (2.00–3.71)	0.25 (0.12–0.49)
	Validation	92(85–97)	61 (54–68)	57 (52–61)	93 (88–96)	2.37 (1.95–2.87)	0.13 (0.07–0.26)
	All	89(83–94)	64 (59–70)	55 (51–59)	93 (89–95) ^a,c^	2.51(2.13–2.96)	0.37(0.28–0.48)
Basophils ≤ 17 cells/mm^3^ and Eosinophils ≤ 10 cells/mm^3^	Inception	56 (40–72)	87 (79–93)	61 (47–73)	85 (80–89)	4.30 (2.50–4.71)	0.50 (0.35–0.72)
	Validation	66 (56–75)	80 (73–86)	65 (57–72)	81 (76–85)	3.30 (2.39–4.56)	0.42 (0.32–0.56)
	All	63 (55–71)	83 (78–87)	64 (57–70) ^a^	82 (79–85) ^d^	3.65 (2.76–4.83)	0.45 (0.36–0.56)
Basophils ≤ 17 cells/mm^3^ and/or Eosinophils ≤ 10 cells/mm^3^	Inception	95 (84–99)	67 (58–75)	51 (44–57)	98 (91–99)	2.88 (2.20–3.77)	0.07 (0.02–0.28)
	Validation	99 (95–100)	53 (45–60)	54 (50–58)	99 (93–100)	2.10 (1.79–2.45)	0.02 (0.00–0.13)
	All	98 (94–100)	58 (53–64)	53 (50–56) ^a^	98 (94–99) ^b,c,d^	2.35 (2.05–2.69)	0.04 (0.01–0.11)

PPV, positive predictive value; NPV, negative predictive value; PLR, positive likelihood ratio value; NLR, negative likelihood ratio. The diagnostic performance of the different tests was compared in the combined cohort (All). Sensitivity and specificity of all tests are significantly different (*p* ≤ 0.001) according the McNemar test. PPV and NPV were compared using the “N-1” Chi-squared test; a,b,c or d, values sharing the same superscript letter were significantly different (*p* < 0.05).

**Table 4 jcm-10-03481-t004:** Diagnostic Performance of B-lines on lung ultrasound alone or combined with eosinophils and basophils below cutoff in the diagnosis of COVID-19 pneumonia.

	Sensitivity	Specificity	PPV	NPV	PLR	NLR
B-lines	86(77–92) ^a,b,c,d^	44(36–51) ^a,b,c,d^	44(41–48) ^a,b,c,d^	85(78–91) ^a^	1.52(1.30–1.78)	0.33(0.19–0.56)
Basophils ≤ 17 cells/mm^3^ and/or Eosinophils ≤ 10 cells/mm^3^	99(94–100) ^a,e,f^	58(50–65) ^a,e^	55(51–59) ^a,e^	99(93–100) ^a,b,c^	2.33(1.95–2.77)	0.02(0.00–0.13)
Basophils ≤ 17 cells/mm^3^ and Eosinophils ≤ 10 cells/mm^3^	69(59–79) ^b,e,g^	79(73–85) ^b,f^	64(56–71) ^b^	83(78–87) ^b^	3.35(2.43–4.62)	0.39(0.28–0.53)
B-lines and Basophils ≤ 17 cells/mm^3^ and Eosinophils ≤ 10 cells/mm^3^	60(50–71) ^c,f,h^	88(82–92) ^c^	72(63–80) ^c,e,f^	81(77–85) ^c^	5.01(3.24–7.73)	0.45(0.35–0.58)
B-lines and/or Basophils ≤ 17 cells/mm^3^ and/or Eosinophils ≤ 10 cells/mm^3^	100(96–100) ^d,g,h^	57(50–65) ^d,e,f^	55(51–59) ^d,f^	100	2.35(1.98–2.79)	0

Data of patients from the inception and validation cohorts who underwent lung ultrasound (91/174, positive/negative patients). Sensitivity and specificity were compared using the McNemar test. PPV and NPV were compared using the “N-1” Chi-squared test; a,b,c,d,e,f,g or h, values sharing the same superscript letter were significantly different (*p* < 0.05).

## Data Availability

The data presented in this study are available on request from the corresponding author. The data are not publicly available due to privacy and ethical reasons.

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
