# Peer review of "Ruling Out Coronavirus Disease 2019 in Patients with Pneumonia: The Role of Blood Cell Count and Lung Ultrasound"

_jcm, 2021, doi:10.3390/jcm10163481_

Round 1

Reviewer 1 Report

As a clinician treating COVID-19 patients and coordinator of a COVID-Team since March 2020, I understand the importance of a sensitive and fast screening tool to rule out COVID-19 in patients with respiratory infection.
The authors explore the combination of low basophils and eosinophils levels with B-lines in lung US as screening of COVID-19 in patients respiratory infections while the NAAT results are pending. They find that basophil and eosinophil count above the cutoff may rule out COVID-19 in patients with respiratory infection with a high sensitivy, which is higher when it is combined with lung US.

The introduction shows an excellent review of a long and widely reviewed topic in recent months. The discussion is concise and focused on the important points of the article. Main limitations have been commented on in the discussion. I would like to suggest that, although it is possible that the difference in mortality between the inception and validation cohort may be due to the different mortality of the waves, as the authors point out in the text, a more detailed description of the sample in table 1 with a comorbidity scale such as Charlson or McCabe, and a severity on admission scale, such as SOFA, CURB-65 or PSI, would help clarify the situation.
Eosinophil count have been considered as independent severity factor in some studies (Mateos-González, J Clin Med 2021; Huang, Medicine 2021). Have the authors considered analyzing this variable to explain the difference in mortality between the inception cohort and the validation cohort? 

Specific comments:
- page 2 line 95 Who excluded the patients? Was the clinical records review done by one or more authors? In the event of a tie, how was the exclusion handled? 

- page 2 line 149 greater or higher?

- page 13 line 269 pandemic surge or wave?

- Table 4: The B-lines and/or Basophils and/or Eosinophils count was mentioned in page 8 line 199, but it was not included in table 4.
- Table 4: Variables Basophils and Eosinophils, as well as Basophils and/or Eosinophils were already included in table 3. It is necessary to duplicate these results? 

Author Response

RESPONSE TO REVIEWER #1

The manuscript was proofread by an English-speaking expert in medicine.

As a clinician treating COVID-19 patients and coordinator of a COVID-Team since March 2020, I understand the importance of a sensitive and fast screening tool to rule out COVID-19 in patients with respiratory infection. The authors explore the combination of low basophils and eosinophils levels with B-lines in lung US as screening of COVID-19 in patients respiratory infections while the NAAT results are pending. They find that basophil and eosinophil count above the cutoff may rule out COVID-19 in patients with respiratory infection with a high sensitivy, which is higher when it is combined with lung US.

Thanks for your kind comments.

The introduction shows an excellent review of a long and widely reviewed topic in recent months. The discussion is concise and focused on the important points of the article. Main limitations have been commented on in the discussion. I would like to suggest that, although it is possible that the difference in mortality between the inception and validation cohort may be due to the different mortality of the waves, as the authors point out in the text, a more detailed description of the sample in table 1 with a comorbidity scale such as Charlson or McCabe, and a severity on admission scale, such as SOFA, CURB-65 or PSI, would help clarify the situation.

Table 1 has been expanded.  We have calculated the Charlson Comorbidity Index and the CURB-65 score. We have included few additional selected laboratory indexes: platelets, hemoglobin, creatinine, and bilirubin. We have modified the METHODS (page 2, line 133-136) and the RESULTS (page 4, line 184-186)  to include these findings. We added REFERENCE #27 and#28. We have modified the DISCUSSION to suggest that the difference in mortality between the inception and validation was mainly due to difference in comorbidities and severity on admission in addition to potential different mortality of the two  waves.

Eosinophil count have been considered as independent severity factor in some studies (Mateos-González, J Clin Med 2021; Huang, Medicine 2021). Have the authors considered analyzing this variable to explain the difference in mortality between the inception cohort and the validation cohort? 

We agree with the reviewer that eosinophil count is an independent severity factor. We underlined this aspect in THE INTRODUCTION (page 2. line.77-78.) and added references #23.. and #24... (Mateos-González, J Clin Med 2021; Huang, Medicine 2021). Nonetheless, in our study, differences in eosinophils cannot account for a higher mortality in the inception cohort because the counts were higher (see Table 1).

Specific comments:

- page 2 line 95 Who excluded the patients? Was the clinical records review done by one or more authors? In the event of a tie, how was the exclusion handled? 

We added the following paragraph to the METHODS (page 3,  line 103-107 ): “Two independent physicians (Gianni Biolo and Pierandrea Vinci) retrospectively evaluated all available records from clinical workup and established the diagnosis of CAP or non-pneumonia disease. When the two physicians were in agreement, a definitive diagnosis was reached. In cases of disagreement, a third independent physician (Filippo Mearelli) established the final diagnosis”. We did not describe this procedure in the first version of the manuscript because disagreement only occurred in two cases.

- page 2 line 149 greater or higher?

we have replaced the word “greater” with the word “higher” (page 5 line 190)

- page 13 line 269 pandemic surge or wave?

we have replaced the word “surge” with the word “wave” throughout the manuscript

- Table 4: The B-lines and/or Basophils and/or Eosinophils count was mentioned in page 8 line 199, but it was not included in table 4.

We have included the results relative to B-lines and/or Basophils and/or Eosinophils in table 4. Statistical results have been added.

- Table 4: Variables Basophils and Eosinophils, as well as Basophils and/or Eosinophils were already included in table 3. It is necessary to duplicate these results? 

The data reported in table 4 refer to the subgroup of patients undergoing LUS, while the data reported in table 3 refer to the entire population. In table 4 we repeated again the results related to eosinophils plus basophils and those related to eosinophils and / or basophils to allow direct statistical evaluation of the contribution of LUS to diagnostic accuracy.

We thank the Reviewer. We feel that the manuscript has improved.

Reviewer 2 Report

  1. Interesting article, well written. This work presents interesting results in the emergency area and can improve quick patient triage within the emergency department, wich is an absolute necessity during COVID-19 pandemic.  
  2. Precise and well conducted statistical analysis.
  3. Weakness: retrospective study with a relatively small amount of data. We cannot say this article is completely original as the link between low eosinophils count and severity of COVID 19 infection has already been study. The association with coagulation disorders has been studied and demonstrated. Moreover, rapid NAAT and low-dose chest CT can now rapidly confirm or rule out COVID 19 infection with great sensitivity and specificity.

Exclusion of severe COVID 19 infection and asymptomatic patients.

However, this work provides new data (inclusion of LUS in particular) and corroborates other recent studies.

  1. Introduction: line 59 à sentence is not completely true. Actually LUS can’t be actually described as the first-line imaging technique in COVID 19 screening and low-dose chest CT allows quantification, high sensitivity, differential diagnosis, with important dose reduction without iodine contrast agent injection..Sentence should be modified.
  2. Material and Methods: mine 98 à LUS examination should be described in details. Was there a double check for images validation, even later after the sonography exam?
  3. Line 104 à reference needed
  4. Table 1 à Validation cohort –eosinophils count of the overall dataset is « 0 (0 – 70) » ?

Présentation could be improved with dfferenciation between clinical and biological values and clear differenciation between inception and validation cohort.

  1. Results à Clinical data in both groups are needed. In this context, some data are important: what were others causes of hospitalization of COVID 19 was eliminate? Underlying respiratory diseases, diabetes or cardiac diseases? Platelets count would have been interesting
  2. Figure 1 àIf possible, authors could include the number of patient who underwent LUS examinations into the flow chart to better understand the overall study.
  3. Results à Clinical data in both groups are needed. Can authors provide the delay between diagnosis and the beginning of symptoms? A statiscally significant difference between groups could lead to bias in the leukocytes count and LUS findings.
  4. Resultsà mortality at 30 days in the inception cohort seems high for non severe COVID 19 infection. Can authors dicuss or explain these results?
  5. Figure 2 à AUC values could be added onto each graph.
  6. Results – line 177 : authors present a 36% and 7% greater sensitivity for the basophils and eosinophils below the cutoffs than in basophils and eosinophils respectively. To which column of the table 3 does  it refer to ? Are p values available for these comparisons?
  7. Table 3 : NPV = Negative Predictive Value”
  8. Discussion à concise, clear.
  9. Discussion à line 224-226: should appear in results section. (p value?)

Author Response

RESPONSE TO REVIEWER #2

The manuscript was proofread by an English-speaking expert in medicine.

  1. Interesting article, well written. This work presents interesting results in the emergency area and can improve quick patient triage within the emergency department, wich is an absolute necessity during COVID-19 pandemic.
  2. Precise and well conducted statistical analysis.
  3. Weakness: retrospective study with a relatively small amount of data. We cannot say this article is completely original as the link between low eosinophils count and severity of COVID 19 infection has already been study. The association with coagulation disorders has been studied and demonstrated. Moreover, rapid NAAT and low-dose chest CT can now rapidly confirm or rule out COVID 19 infection with great sensitivity and specificity.

Thanks for your kind comments.

We have added a sentence to the INTRODUCTION (Page 2 Line 61-64) and REFERENCE #6 AND #11 to describe the efficiency of low-dose chest CT in combination with rapid NAAT to rapidly confirm or rule out COVID 19 infection with great sensitivity and specificity.

Exclusion of severe COVID 19 infection and asymptomatic patients.

However, this work provides new data (inclusion of LUS in particular) and corroborates other recent studies.

  1. Introduction: line 59 à sentence is not completely true. Actually LUS can’t be actually described as the first-line imaging technique in COVID 19 screening and low-dose chest CT allows quantification, high sensitivity, differential diagnosis, with important dose reduction without iodine contrast agent injection. Sentence should be modified.

We have modified the sentence in agreement with your suggestion (page 2 line 61-64 and line 65).

  1. Material and Methods: mine 98 à LUS examination should be described in details. Was there a double check for images validation, even later after the sonography exam?

We have added a description of LUS examination in METHODS (page 3 line 111-126) and REFERENCE #26.

  1. Line 104 à reference needed

Reference #13.

  1. Table 1 à Validation cohort –eosinophils count of the overall dataset is « 0 (0 – 70) » ?

Yes, IT IS correct

Présentation could be improved with dfferenciation between clinical and biological values and clear differenciation between inception and validation cohort.

  1. Results à Clinical data in both groups are needed. In this context, some data are important: what were others causes of hospitalization of COVID 19 was eliminate? Underlying respiratory diseases, diabetes or cardiac diseases? Platelets count would have been interesting.
  • We have described in the RESULTS the causes of hospitalization of the 234 patients who were excluded from the analysis in the inception and validation cohorts (page 4 line 169-179).
  • Comorbidities were assessed by calculation of the Charlson comorbidity index.
  • Table 1 has been expanded.  We have calculated the Charlson Comorbidity Index and the CURB-65 score. We have included few additional selected laboratory indexes: platelets, hemoglobin, creatinine, and bilirubin. We have modified the METHODS (page 2, line 133-136) and the RESULTS (page 4, line 184-186)  to include these findings. We added REFERENCE #27 and#28.

  1. Figure 1 àIf possible, authors could include the number of patient who underwent LUS examinations into the flow chart to better understand the overall study.

We included the number of patients who underwent LUS in Figure 1.

  1. Results à Clinical data in both groups are needed. Can authors provide the delay between diagnosis and the beginning of symptoms? A statiscally significant difference between groups could lead to bias in the leukocytes count and LUS findings.
  • Table 1 has been expanded.  We have calculated the Charlson Comorbidity Index and the CURB-65 score. We have included few additional selected laboratory indexes: platelets, hemoglobin, creatinine, and bilirubin. We have modified the METHODS (page 2, line 133-136) and the RESULTS (page 4, line 184-186)  to include these findings. We added REFERENCE #27 and#28.
  • Clinical records inaccurately reported the time from onset of symptoms to hospitalization in less than 50% of COVID-19 positive patients in the validation cohort as follows: median (IQR) 6 (3-10) days. No information was collected in the inception cohort. We think this information is too imprecise to be reported in the manuscript.

  1. Resultsà mortality at 30 days in the inception cohort seems high for non severe COVID 19 infection. Can authors dicuss or explain these results?

We have modified the DISCUSSION to suggest that the difference in mortality between the inception and validation was mainly due to difference in comorbidities and severity on admission, in addition to potential different mortality of the two  waves (page 12 line 314-327).

We added two scores of comorbidity and severity in table 1, highlighting the severe conditions of many of these patients. A comment on mortality, moreover, has been reported in Discussion (lines 497-520)

  1. Figure 2 à AUC values could be added onto each graph.

AUC values are reported in Table 2. This is described in the legend of Figure 2.

  1. Results – line 177 : authors present a 36% and 7% greater sensitivity for the basophils and eosinophils below the cutoffs than in basophils and eosinophils respectively. To which column of the table 3 does it refer to ? Are p values available for these comparisons?

In Table 3 The diagnostic performance of the different tests was compared in combined cohorts (All). Results of statistics are reported in legend of Table 3. We modified the text of RESULTS (page 13 line 218-222) to make this point clear.

  1. Table 3 : NPV = Negative Predictive Value”

Thank you. We have corrected the typo.

  1. Discussion à concise, clear.
  2. Thank you.

  1. Discussion à line 224-226: should appear in results section. (p value?)

We moved the sentence (“Among patients with COVID-19, 82% had zero eosinophils at presentation, compared to 29% of non COVID-19 patients (p<0.05). Median basophil count was 68% lower in the COVID-19 as compared to that of non COVID-19 patients (p<0.05).”)  from Discussion to Results (page 5 line 193-195). We added the p value.

We thank the Reviewer. We feel that the manuscript has improved.